# Pan-Cancer Analysis Reveals the Prognostic Potential of the THAP9/THAP9-AS1 Sense–Antisense Gene Pair in Human Cancers

**DOI:** 10.3390/ncrna8040051

**Published:** 2022-07-08

**Authors:** Richa Rashmi, Sharmistha Majumdar

**Affiliations:** Discipline of Biological Engineering, Indian Institute of Technology Gandhinagar, Gandhinagar 382355, India; richa.rashmi@iitgn.ac.in

**Keywords:** THAP9, THAP9-AS1, head-to-head genes, pan-cancer, TCGA, survival, co-expression, guilt-by-association

## Abstract

Human THAP9, which encodes a domesticated transposase of unknown function, and lncRNA THAP9-AS1 (THAP9-antisense1) are arranged head-to-head on opposite DNA strands, forming a sense and antisense gene pair. We predict that there is a bidirectional promoter that potentially regulates the expression of THAP9 and THAP9-AS1. Although both THAP9 and THAP9-AS1 are reported to be involved in various cancers, their correlative roles on each other’s expression has not been explored. We analyzed the expression levels, prognosis, and predicted biological functions of the two genes across different cancer datasets (TCGA, GTEx). We observed that although the expression levels of the two genes, THAP9 and THAP9-AS1, varied in different tumors, the expression of the gene pair was strongly correlated with patient prognosis; higher expression of the gene pair was usually linked to poor overall and disease-free survival. Thus, THAP9 and THAP9-AS1 may serve as potential clinical biomarkers of tumor prognosis. Further, we performed a gene co-expression analysis (using WGCNA) followed by a differential gene correlation analysis (DGCA) across 22 cancers to identify genes that share the expression pattern of THAP9 and THAP9-AS1. Interestingly, in both normal and cancer samples, THAP9 and THAP9-AS1 often co-express; moreover, their expression is positively correlated in each cancer type, suggesting the coordinated regulation of this H2H gene pair.

## 1. Introduction

If the 5′ ends of two genes are adjacent to one another on opposite DNA strands, and the two genes are transcribed divergently, they are called head-to-head genes. The region between the transcription start sites (TSS) of head-to-head genes can be identified as a putative bidirectional promoter. Eukaryotic genes are sometimes organized in a head-to-head architecture, sharing a bidirectional promoter region for regulating the expression of the two genes [1]. Genome-wide analyses have shown that more than 10% of human genes are arranged in a bidirectional head-to-head architecture with their TSSs located <1 kb apart [2,3,4].

The transcriptional regulation of genes that share a bidirectional promoter is complex. They can be positively correlated such as human collagen genes COL4A1 and COL4A2 [5,6], negatively correlated such as mouse TO/KF [7], or they can show tissue-specific or condition-specific correlations such as human HSP60 and HSP10, which are coordinated to respond to induction signals [8]. Interestingly, several recent genome-wide studies have reported that the sense gene expression positively correlates with the expression of the corresponding antisense gene in the same tissues or cells. For instance, the expression of [9] 38% of annotated antisense RNA transcripts positively correlated with sense gene expression in 376 cancer samples comprising nine tissue types. Moreover, several bidirectional gene pairs are associated with human diseases, such as BRCA1/NBR2 [10], ATM/NPAT [11], DHFR/MSH3 [12], and SERPINII/PDCD10 [13].

The THAP9 and THAP9-AS1 genes are a putative bidirectional gene pair; their TSSs are located 166 bases apart, and they are arranged in a head-to-head or divergent manner on opposite DNA strands. THAP9 is a domesticated human DNA transposase, homologous to the widely studied *Drosophila* P-element transposase [14]. The THAP9 protein shares 40% similarity to the P-element transposase and probably does not transpose in vivo due to the absence of terminal inverted repeats and target site duplications. Despite being domesticated, it has retained its catalytic activity [14,15]. Additionally, hTHAP9 belongs to the THAP (Thanatos-associated protein) protein family in humans, containing twelve members (hTHAP0-hTHAP11). All human THAP proteins are characterized by an amino terminal DNA-binding domain called the THAP domain, which is typically 80–90 amino acid residues long and possesses a C2CH-type zinc finger [16,17].

Many THAP family proteins are known to be involved in human diseases. THAP1 has been associated with DYT6 dystonia (a hereditary movement disorder involving sustained involuntary muscle contractions) [18]. Regulation of THAP5 by Omi/HtrA2 has been linked to cell cycle control and apoptosis in cardiomyocytes [19]. THAP1 plays a role in apoptosis by facilitating programmed cell death with the help of the transcription repressor protein Par-4 [20]. The LRRC49/THAP10 bidirectional gene pair is reported to have reduced expression in breast cancer [21]. THAP11 is differentially expressed during human colon cancer progression and acts as a cell growth suppressor by negatively regulating the c-Myc pathway in gastric cancer [22].

THAP9-AS1 is a long non-coding RNA gene located on chromosome 4q21.22. It is upregulated in nasopharyngeal carcinoma and breast cancer [23,24]. Further, it was also identified as an oncogenic factor promoting cell growth in pancreatic ductal adenocarcinoma and gastric cancer and plays a role in the apoptosis of spontaneous neutrophils [25,26,27]. THAP9-AS1 knockdown suppresses cell proliferation and enhances apoptosis in esophageal squamous cell carcinoma (ESCC) [28]. A recent study reported that THAP9 and THAP9-AS1 exhibit different gene expression patterns under various stress conditions in the S-phase of the cell cycle. THAP9-AS1 is consistently upregulated under stress, whereas THAP9 exhibits both downregulation and upregulation. Both THAP9 and THAP9-AS1 exhibit periodic gene expression throughout the S-phase, which is characteristic of cell-cycle-regulated genes [29]. Nevertheless, little is known about the biochemical and biological functions of the products encoded by THAP9 and THAP9-AS1 or their combined role in tumorigenesis.

In this study, we were interested in understanding the individual as well as combined roles of THAP9 and THAP9-AS1 in various cancers. We conducted a pan-cancer bioinformatic analysis of THAP9 and THAP9-AS1 focusing on their expression, patient prognosis, and genetic mutations in TCGA [30] and GTEx datasets [31] via TIMER2 [32], GEPIA2 [33] and cBio portal [34]. We used a weighted gene co-expression network analysis (WGCNA) [35] and differential gene correlation analysis (DGCA) [36] to explore the gene expression datasets from TCGA for correlations in expression between THAP9 and THAP9-AS1 and their correlation with other genes. Gene Ontology (GO) and KEGG pathway enrichment analyses were performed to identify the primary biological functions linked to the genes that share the THAP9 and THAP9-AS1 clusters in various tumor and normal samples. Our findings identified genetic mutations and indicated statistical correlations between the expression of THAP9 and THAP9-AS1 and clinical prognosis and several cancer-related pathways, which suggests that the gene pair can serve as a potential prognostic cancer biomarker.

## 2. Results

### 2.1. Characterization of THAP9/THAP9-AS1 Promoter

THAP9 is a domesticated transposase [14] and THAP9-AS1 (THAP9-antisense1, ENST00000504520) is a lncRNA [37]. Together they form a sense and antisense gene pair organized in a “head-to-head” (H2H) orientation (Figure 1a). The sequence between an H2H gene pair, i.e., intra-H2H pair, can act as a bidirectional promoter [3]. It has been reported that bidirectional promoters for H2H gene pairs may regulate the expression of the two genes [1] under specific conditions (e.g., disease).

Before investigating the independent and combined role of THAP9/THAP9-AS1 in tumorigenesis, we explored the possibility of a putative bidirectional promoter for the gene pair that may be involved in the regulation of their expression. Thus, we performed an *in silico* analysis of the genomic region spanning their predicted transcriptional start sites (TSS).

According to EPDnew [38], the TSS of THAP9 is located at position 82900735 (Appendix A) on the sense strand of chromosome 4, while two TSSs were predicted for THAP9-AS1 at positions 82900569 and 82900944 (Appendix A), both located on the antisense strand of chromosome 4. Thus, the predicted intergenic region between THAP9 and THAP9-AS1 is 166 bp (non-overlapping if THAP9-AS1 TSS is at position 82900569) or 209 bp (overlapping if THAP9-AS1 TSS is at position 82900944). In both cases, the THAP9-THAP9-AS1 sense antisense pair follows the head-to-head architecture (Figure 1a). We assumed that the predicted bidirectional promoter region was located in the region spanning −250 to +250 relative to the predicted THAP9 TSS and −400 to +100 assuming position 82900569 as the TSS for THAP9-AS1 (the selected sequence includes both TSSs for THAP9-AS1), and downloaded the corresponding sequence from EPDnew for further characterization (Appendix A).

Most bidirectional promoters are characterized by the presence of CpG islands, which are regions that are usually devoid of DNA methylation and have a higher G+C content [39]. Thus, we examined the presence of CpG islands (CGIs) in the THAP9/THAP9-AS1 predicted bidirectional promoter region. The analysis and visualization of the selected promoter sequences for each gene, i.e., THAP9 and THAP9-AS1 using EMBOSS CPGplot (https://www.ebi.ac.uk/Tools/seqstats/emboss_cpgplot/ accessed on 15 September 2021), established that they individually fulfilled the criteria for CGIs [40] and showed a GC content (Percent C + Percent G) > 50.00 and observed/expected CpG ratio > 0.60 (Appendix A). We then looked for already annotated CGIs around the THAP9/THAP-AS1 predicted bidirectional promoter region using the UCSC Genome Browser (http://genome.ucsc.edu/ accessed on 21 September 2021). We observed a CGI located between positions 82900535 and 82900912 on chromosome 4 overlapping with the putative promoter (Figure 1b). It is tempting to speculate that differential DNA methylation of this CGI may influence the bidirectional gene expression of THAP9/THAP-AS1.

In addition to CGI, bidirectional promoters are often enriched with specific histone marks. Thus, we decided to look for the already annotated histone mark profile in the THAP9/THAP9-AS1 bidirectional promoter region using ENCODE (Encyclopedia of DNA Elements) [41] (datasets used—EH38E3592191, EH38E3592192), which revealed the presence of bimodal peaks of transcriptionally active histone modifications namely H3K4Me1 (associated with enhancers and the downstream region of TSS), H3K4Me3 (associated with promoters that are active or poised to be activated), and H3K27Ac (associated with enhanced transcription by blocking the spread of the repressive histone mark H3K27Me3). (Figure 2). Moreover, the THAP9/THAP9-AS1 bidirectional promoter was also characterized by DNase I hypersensitivity, which suggests bidirectional transcriptional activity [42]. The 7 cell lines included in this track are GM12878 (lymphoblastoid cell line), H1-hESC (embryonic stem cell line derived from human blastocysts), HSMM (human skeletal muscle myoblasts), HUVEC (human umbilical vein endothelial cells), K562 (human immortalized myelogenous leukemia cell line), NHEK (primary human keratinocytes), and NHLF (human lung fibroblasts).

While there is no consensus on a computational method to predict whether a promoter is bidirectional or not, certain core promoter elements are known to be essential structural features of bidirectional promoters. These elements include the TATA box, CCAAT box, B recognition element (BRE), initiator element (INR), and downstream promoter element (DPE) [44]. The TATA box exists in both unidirectional and bidirectional promoters; however, they are less frequent amongst bidirectional promoters. Bidirectional promoters generally have a higher enrichment of CCAAT boxes and the BRE element compared to unidirectional promoters, while the ratio of DPE to INR remain largely unchanged [44].

Therefore, we submitted the THAP9 and THAP9-AS1 promoter sequences to ElemeNT (https://www.juven-gershonlab.org/resources/element/run/ accessed on 29 September 2021) [45] to predict putative core promoter elements (Figure 2b). The elements identified within the predicted promoter region included BRE, GAGA, mammalian initiator (INR), bridge 1, DPE, BBCABW initiator, human TCT, and TFIIA response element, but not the TATA box (Figure 2b, Appendix A). It is known that in TATA-less promoters, the INR element is the point of transcription initiation [46]. The DPE acts together with INR and is required for binding TFIID [47] and TBP-associated factors (TAFs), particularly TAF6 and TAF9 [48]. Moreover, in the TATA-less promoters, the TFIIB–BRE interaction plays a vital role in assembling the preinitiation complex and transcription initiation [49]. Thus, we note that the selected THAP9/THAP9-AS1 promoter region displays several features (BRE, DPE, INR) characteristic of bidirectional promoters and may be regulated in a coordinated fashion. Thus, the absence of TATA box and presence of other core promoter elements, CGI and histone mark signatures in the region between the TSS of THAP9 and THAP9-AS1, strongly suggest that this region contains a bidirectional promoter.

### 2.2. Analysis of THAP9/THAP9-AS1 Mutations in Various Tumors

The possible roles of THAP9 in disease has not been investigated. No disease-specific mutation has been reported for the THAP9 gene till date. The only reported missense SNP in THAP9 with a global MAF > 0.05 (minor allele frequency) was rs897945 (MAF = 0.3253), making it a candidate for a derived allele (DA). Derived alleles are new alleles, having a population frequency of at least 5%, which are formed by the mutation of ancestral alleles. It has been reported that disease-associated alleles are more likely to be low-frequency derived alleles [50]. In the *hTHAP9* gene, rs897945 yields a G → T nucleotide substitution, leading to a leucine-to-phenylalanine mutation at position 299 on the Tnp_P_element (Pfam ID: PF12017) domain (Figure 3). This SNP may have a role in atopy and allergic rhinitis in a Singaporean Chinese population [51].

To understand the disease association of THAP9 and THAP9-AS1 in more detail, we next studied the prevalence of their genetic alterations across various human cancers, using the cBioPortal tool [34] in the “pan-cancer analysis of whole genomes (ICGC/TCGA, Nature 2020)” dataset available at http://www.cbioportal.org accessed on 3 October 2021. As shown in Figure 4a,d, pancreatic cancer patients had the highest alteration frequencies in both THAP9 and THAP9-AS1 (>6%), with “amplification” (i.e., more copies, often focal) being the primary alteration type. Notably, both THAP9 and THAP9-AS1 genes underwent amplification in breast cancer, non-small cell lung cancer, melanoma, embryonal tumor, and bone cancer but underwent “deep deletion” (indicates a deep loss, possibly a homozygous deletion) in uterine endometrioid carcinoma patients. Moreover, for the THAP9 gene, “mutation” appeared as the only form of alteration in all patients with colorectal cancer, mature B-cell Lymphoma, and head and neck Cancer. The sites and types of THAP9 mutations are presented in Figure 4c (Appendix A). The median months survival time was 37.73 for the altered THAP9 group (Figure 4b, 49.61 for reference group). However, it is to be noted that in Figure 4b,e, the *p*-values for both THAP9 and THAP9-AS1 are not less than 0.05, rather the value is 0.2 for THAP9 and 0.5 for THAP9-AS1, which implies that the alterations may not reflect a significant impact on the poor overall survival and prognosis in cancer. This also correlates with the alteration frequency displayed in Figure 4a,d; none of the alterations show more than a 6% frequency.

### 2.3. Difference between THAP9/THAP9-AS1 Expression in Several Cancers

To compare the expression levels of THAP9 and THAP9-AS1 genes between tumor and normal samples, we analyzed their expression levels across various cancer types using TCGA [30] and GTEx [31] datasets via TIMER2.0 [32] and GEPIA2 [33].

TIMER2: We first used TIMER2.0 to evaluate the expression levels of THAP9 between primary tumor and normal samples using TCGA database. We found that THAP9 expression levels in tumor tissues of CHOL (*p* < 0.001), COAD (*p* < 0.001), ESCA (*p* < 0.01), LIHC (*p* < 0.001), LUSC (*p* < 0.001), LUAD (*p* < 0.05), and STAD (*p* < 0.001) were higher than the corresponding normal tissue (Figure 5). On the contrary, THAP9 expression levels in tumor tissues of KIRC (*p* < 0.001), KIRP (*p* < 0.001), PRAD (*p* < 0.05), THCA (*p* < 0.001), and UCEC (*p* < 0.001) were lower than the corresponding normal tissue (Figure 5). We could not perform a similar comparison in ACC, DLBC, LAML, LGG, MESO, OV, SARC, SKCM, TGCT, UCS, or UVM because they lack normal samples in TCGA database. A similar analysis could not be conducted for THAP9-AS1, since TIMER2.0 does not give any gene expression profile for THAP9-AS1.

GEPIA2: Since the TIMER2.0 database did not have the gene expression profile for THAP9-AS1, to get a broader understanding of the gene expression profiles of the two genes we used GEPIA2. It analyzes genes using both TCGA and GTEx datasets. It helped us obtain the differential gene expression profiles of THAP9 and THAP9-AS1 across 31 cancers (pan-cancer gene expression profiles of THAP9 and THAP9-AS1 in Appendix A, respectively). We observed that THAP9 expression was downregulated in TGCT (*p* < 0.01) and was upregulated in THYM (*p* < 0.01), (Figure 6a,b). However, THAP9-AS1 showed downregulation in OV (*p* < 0.01), SKCM (*p* < 0.01), and THCA (*p* < 0.01) (Figure 6c–e) and was upregulated in THYM (*p* < 0.01), PAAD (*p* < 0.01), DLBC (*p* < 0.01), and CHOL (*p* < 0.01) (Figure 6f–i).

Combining the results from the two methods, we observed that THAP9 and THAP9-AS1 expression levels were coordinately upregulated in CHOL and THYM and were coordinately downregulated in THCA compared with the corresponding normal samples.

Head-to-head genes are often coregulated by bidirectional promoters, although there have been reports of conditional regulation of bidirectional gene pairs as well. For example, some gene pairs such as murine RanBP1/Htf9-c are coregulated only in a common window of the cell cycle [7,52]. On the other hand, human HSP60/HSP10 displays coordinated expression in response to induction signals [8]. We have previously reported that THAP9 and THAP9-AS1 exhibit different gene expression patterns under diverse stress conditions in the S-phase of the cell cycle. THAP9-AS1 is consistently upregulated under stress, whereas THAP9 exhibits both downregulation and upregulation [29].

Thus, given the above, it is possible that THAP9 and THAP9-AS1 show diverse expression patterns in different cancers. The differential expression of THAP9 and THAP9-AS1 in different tumor types suggests that the two genes may have tumor-specific regulatory mechanisms.

### 2.4. Prognostic Analysis of THAP9 and THAP9-AS1

We used the datasets from TCGA and GTEx via GEPIA2 to investigate the correlation of THAP9 and THAP9-AS1 expression with patients’ prognoses across different tumor types. The survival heat map of hazard ratio (HR) values for overall and disease-free survival (Figure 7a and Figure 8a) shows the prognostic impacts of THAP9 and THAP9-AS1 in multiple cancer types. A poor prognosis and poor overall survival were linked to the upregulation of THAP9 expression in LGG and STAD and its downregulation in HNSC and KIRC (Figure 7b). Moreover, poor DFS (disease-free survival) was linked with upregulated THAP9 expression in BLCA and CESC and its downregulated expression in KIRC and THYM (Figure 8b). Similarly, in the case of THAP9-AS1, its upregulation was linked to a poor prognosis and poor overall survival in ACC, LGG, PRAD, SARC, and THCA (Figure 7c) and poor DFS in ACC, KICH, and MESO, while its downregulated expression was linked to poor DFS in KIRC (Figure 8c).

These findings suggest that *THAP9-* and *THAP9-AS1* related cancer prognoses differ with different cancer types and show much less correlation with the cancer types in which the two genes are differentially expressed. Therefore, the potential of using these genes as a pan-cancer survival indicator is limited.

### 2.5. Understanding the Role of THAP9 and THAP9-AS1 Using Guilt-By-Association Analysis

A GBA (guilt by association) [53] analysis is often used to predict an unknown gene’s function by grouping it with known genes that share its transcriptional behavior. Genes turned on or turned off together under various conditions may be part of the same cellular processes [54]. The exact cellular functions of THAP9 and THAP9-AS1 are unknown. Thus, we decided to compare the expression of THAP9 with THAP9-AS1 and 34125 other genes in 9571 tumor and associated normal samples from 22 human cancers fetched from TCGA [using HTSeq count datasets from TCGA (Appendix A), excluding cancer types with less than 3 normal samples (rows highlighted in red)]. The gene co-expression network for THAP9 and THAP9-AS1 was constructed using WGCNA followed by a differential gene correlation analysis using DGCA. We also investigated Gene Ontology and KEGG pathways (Appendix A) for the genes co-expressed with the two genes and the genes differentially correlated across normal vs. tumor samples (Appendix A). An analysis of the functions of genes co-expressed with THAP9 and THAP9-AS1 may provide insights into their possible physiological roles.

#### 2.5.1. Gene Co-Expression Analysis

To identify the genes that are co-expressed with THAP9 and THAP9-AS1 in normal vs. tumor samples in each cancer type, we utilized the WGCNA R package [35] to build a weighted co-expression network for the two genes. The samples of each tumor and normal pair were clustered to identify the gene modules representing genes co-expressed with THAP9 and THAP9-AS1. To represent the most frequently co-expressed genes, we selected the top 20 genes co-expressed with THAP9 (Appendix A) and THAP9-AS1 (Appendix A), combining all tumors and normal samples across the cancer types (Figure 9,).

In normal samples (within BRCA, KIRC, STAD, and THCA datasets), THAP9 and THAP9-AS1 belong to the same gene cluster, suggesting their coregulation (Appendix A). Some genes that are overrepresented in these clusters are GPBP1, API5, PIK2C3, GOSR1 and PRPF40A. GPBP1, also known as Vasculin, is a promoter binding protein that is reported to have roles in atherosclerosis [55], hypertension, hypercholesterolemia [56], and Alzheimer’s disease [57,58]. AP15 prevents apoptosis in the absence of growth factor [59,60,61,62], while the BECN1-PIK3C3 complex plays a crucial role in autophagy [63]. The GOSR1 protein is responsible for cellular trafficking [64] and is frequently upregulated in esophageal squamous cell carcinoma tissues [65]. PRPF40A is associated with pre-mRNA splicing [66,67], genetic diseases such as Rett syndrome [68], Huntington’s disease [69], and cancers [70] such as lung cancer [71] and pancreatic ductal adenocarcinoma [72].

Similarly, in tumor samples, the two genes are part of the same gene cluster in HNSC, LUAD, STAD, and UCEC. Genes overrepresented in these clusters include YTHDC1, SRSF10, BLCAF1, MFSD8, and ABHD18. YTHDC1 is a nuclear protein involved in splicing cancer-causing transcripts [73] and plays a regulatory role in several cancers such as breast and prostate cancer [74,75,76,77]. SRSF10 is an SR protein and splicing regulator that activates splicing when phosphorylated and inhibits splicing when dephosphorylated [78,79]. Bclaf1 is a tumor suppressor gene [80] involved in T-cell activation [81], repairing DNA damage [82,83], and pre-mRNA splicing [84], with a regulatory role in colon cancer [79]. Alterations in MFSD8 have been associated with a neurodegenerative disorder called vLINCL, which causes seizures, progressive mental and motor deterioration, myoclonus, visual failure, and premature death [85,86,87,88,89]. The ABHD18 protein is a genetic marker for hepatocellular carcinoma (HCC) in Asian populations [90].

Next, we set to identify the functional association of the THAP9 and THAP9-AS1 co-expression modules. We used ‘ShinyGO’, which performs an in-depth analysis of gene lists that includes a graphical visualization of enrichment, pathway, gene characteristic, and protein interactions [91]. The noteworthy pathways from GO analysis for THAP9 and THAP9-AS1 are visualized in Figure 10 and Figure 11, respectively (Appendix A).

*THAP9:* As presented in Figure 10 (Appendix A), in normal individuals, genes co-expressed with THAP9 were involved in RNA biosynthesis and organelle organization (Figure 10a). The co-expressed genes in both normal and tumor samples were markedly enriched in the nucleoplasm and nuclear lumen (Figure 10b) and were significantly involved in binding nucleic acids (especially DNA) (Figure 10c); this is interesting as THAP9 also appears to localize in the nucleus [92] and possibly binds DNA via an amino terminal DNA-binding THAP domain [17]. We also observed the enrichment of several KEGG pathways associated with (Figure 10d) Herpes simplex virus 1 infection (normal and tumor), as well as neurodegenerative disorders such as Alzheimer’s disease, Parkinson’s disease, and Huntington’s disease (normal samples).

*THAP9-AS1:* Similarly, Figure 11 (Appendix A) shows that genes co-expressed with THAP9-AS1 in both normal and tumor samples were markedly enriched in the nucleoplasm and nuclear lumen (Figure 11b) and associated with herpes simplex virus 1 infection (Figure 11d). In contrast, it appeared that the co-expressed genes were significantly enriched in cilia only in the tumor samples and involved in RNA binding in normal samples.

#### 2.5.2. Differential Gene Correlation Analysis

A differential gene correlation or co-expression analysis can identify biologically important differentially correlated genes that cannot be detected using a regular gene co-expression analysis or differential gene expression analyses. It is suggested that if there is a change in the correlation between the expression of two genes under certain conditions (i.e., they are differentially correlated), they possibly regulate or are regulated by the condition [93,94,95]. Many studies have used differential correlation analyses to identify genes underlying differences between healthy and diseased samples or between different tissues, cell types, or species [96,97,98,99]. Genes that are functionally related tend to have similar expression profiles; therefore, a differential gene correlation analysis that can compare the expression correlation of THAP9 and THAP9-AS1 with other genes in normal vs. tumor samples can give us insight into biological processes and molecular pathways that distinctly involve the two genes in the two conditions.

In this study, we used a DGCA to identify the genes differentially correlated with THAP9 (Appendix A). and THAP9-AS1(Appendix A) under various tumor vs. paired normal conditions. We used the RNA-seq HTSeq count dataset of 22 cancer and paired normal samples from TCGA. A pan-cancer analysis of head-to-head gene pairs [100] reported that these gene pairs show significantly stronger positive correlations in tumor compared to normal samples, regardless of tumor types. Moreover, bidirectional promoters are known to regulate the expression of many cancer-related genes such as BRCA1 and TP53 [101,102]. Thus, we calculated the correlation between the THAP9 and THAP9-AS1 H2H genes.

Interestingly, when we compared the expression patterns of THAP9 and THAP9-AS1 in normal and tumor samples, they always showed a positive correlationin each cancer type, suggesting their coordinated regulation (Appendix A). It is noteworthy that we have also predicted a bidirectional promoter region between the two genes (Figure 1).

We then calculated the differences in Spearman correlations for all genes with THAP9 and THAP9-AS1 to identify the genes differentially correlated with THAP9 and THAP9-AS1 between the tumor and the paired normal samples in each cancer type. Further, we measured the Gene Ontology (GO) enrichment of the genes differentially correlated with THAP9 and THAP9-AS1 with a gain and loss of correlation in tumor vs. normal samples.

THAP9: Looking at the combined results from all the cancers (Figure 12, results for each cancer separately in Appendix A), genes that lost correlation with THAP9 were enriched in nuclear chromatin (Figure 12b) and often involved in processes such as DNA-mediated transposition, negative regulation of gene expression, and ion channel activity (Figure 12a,c).

*THAP9-AS1:* In all cancer samples (Figure 13, results for each cancer separately in Appendix A), genes that gained in correlation with THAP-AS1 were enriched in immune system processes (Figure 13a,c). Moreover, genes that showed a correlated expression (gain or loss) with THAP9-AS1 were enriched in the plasma membrane and cytoplasm.

## 3. Discussion

This study investigated the pan-cancer expression patterns of THAP9 and THAP9-AS1, which are a pair of sense–antisense genes that occur in a “head-to-head” orientation on chromosome 4q21. The human genome contains numerous pairs of genes with similar “head-to-head” orientations with transcription start sites separated by less than 1 kb [2]. Several of these gene pairs are regulated by a single bidirectional promoter [103]. Bidirectional promoters typically have high GC contents, frequently lack TATA boxes, and are often conserved among mouse orthologs [44,104]. Interestingly, these structural features are also present in the predicted bidirectional promoter region of the THAP9/THAP9-AS1 gene pair.

Bidirectional promoters may regulate the coordinated expression of two genes (within a gene pair) that have complementary roles [105] and help maintain stoichiometric quantities of each gene’s expression [103]. They are also responsible for driving the transcription of genes involved in the same cellular pathway or genes that need to be sequentially activated [52,105]. Thus, we decided to investigate whether the expression levels of THAP9 and THAP9-AS1 were correlated and if the gene pair was possibly regulated by their putative bidirectional promoter.

It has been reported that THAP9 is a highly conserved gene, which has been identified in 178 organisms [106]. The human THAP9 gene has 6 isoforms, out of which only one is known to encode for aprotein that is homologous to the *Drosophila* P-element transposase. hTHAP9 belongs to the THAP (Thanatos-associated protein) protein family in humans, containing twelve proteins (hTHAP0-hTHAP11) [17]. Many THAP family proteins are known to be involved in human diseases. THAP1 has been associated with DYT6 dystonia [18], THAP5 and THAP1 have been linked to apoptosis [19,20], the LRRC49/THAP10 bidirectional gene pair is involved in breast cancer [21], and THAP11 has been implicated in colon and gastric cancers [22,107]. THAP9-AS1 (THAP9 antisense) is a newly annotated (by Ensembl) lncRNA coding gene that encodes 12 long non-coding RNAs. Recent reports have suggested that the THAP9-AS1 lncRNA is involved in pancreatic cancer, septic shock, and neutrophil apoptosis [25,26]. However, the roles of THAP9 and THAP9-AS1 across human cancers are not well understood. This study investigated the relationship between THAP9 and THAP9-AS1 expression and their possible roles in tumorigenesis via a pan-cancer analysis of TCGA and GTEx databases.

The gene expression analysis using TIMER2 and GEPIA2 suggested that both over- and under-expression of THAP9 and THAP9-AS1 frequently occurred in various cancers. We observed that THAP9 was upregulated in CHOL, COAD, ESCA, LIHC, LUSC, LUAD, STAD, and THYM and downregulated in KIRC, KIRP, PRAD, THCA, TGCT, and UCEC. On the other hand, THAP9-AS1 was upregulated in CHOL, THYM, DLBC, and PAAD, while it was downregulated in OV, SKCM, and THCA. We also observed that compared with the corresponding normal tissues, THAP9 and THAP9-AS1 expression levels were coordinately upregulated in CHOL and THYM but coordinately downregulated in THCA. Therefore, the independent and coordinated alteration in THAP9 and THAP9-AS1 expression in various cancers indicates that they may have different biological functions in different cancers. Regardless, the aberrant expression levels of the THAP9 and THAP9-AS1 gene pair were associated with a poor prognosis in many types of cancer, which suggested their role as a potential prognostic cancer biomarker. Moreover, we observed that both the THAP9 and THAP9-AS1 genes were mutated (often amplified) in several cancers (TCGA dataset).

Comprehensive gene expression studies can help in predicting gene function. Genes that share an expression pattern, i.e., are turned on or off together under various conditions, may encode proteins that constitute the same multiprotein machine or are involved in a complex, coordinated activity. Characterizing an unknown gene’s function by grouping it with known genes that share its transcriptional behavior is called a “guilt by association analysis (GBA)” [108], which can be simply explained as “a man is known by the company he keeps”. A GBA involves a gene co-expression analysis followed by gene interaction network construction by clustering associations from gene co-expression data. Genes that belong to the same cluster may be involved in common cellular pathways or processes. 

Our GO analysis suggested that many of the genes that were co-expressed with THAP9 were also involved in DNA and metal binding, much like THAP9 homologs like *Drosophila* P-element transposase, which binds DNA via a characteristic zinc-finger-type THAP domain. KEGG pathway analysis of genes co-expressed with THAP9 and THAP9-AS1 demonstrated the enrichment of pathways related to Herpes simplex virus 1 infection as well as several neurodegenerative disorders like Alzheimer’s disease, Parkinson’s disease, and Huntington’s disease. It has been suggested that herpes simplex virus 1 infection may be a causative agent of Alzheimer’s disease [109]. THAP9 has previously been reported to be upregulated (5-fold) in tuberculous meningitis (TBM) patients co-infected with HIV compared to patients with TBM alone [110]. It will be interesting to investigate the possible role of THAP9 in neurological disorders.

Our analysis demonstrates that although THAP9 and THAP9-AS1 show diverse expression patterns in different cancers, their expression in normal and tumor samples was positively correlated in each cancer type. This suggests the coordinated regulation of the two genes (Appendix A). It is tempting to speculate that this coordinated regulation is mediated by the predicted bidirectional promoter region between the two genes (Figure 1). Moreover, the differential expression of THAP9 and THAP9-AS1 in different tumor types suggests that the two genes may have tumor-specific regulatory mechanisms.

## 4. Materials and Methods

### 4.1. Analysis of Promoter Sequence

The promoter sequences were downloaded using EPDnew [38], which is a new section under the well-known Eukaryotic Promoter Database (EPD) (https://epd.epfl.ch accessed on 13 September 2021) [38]. The EPD is an annotated non-redundant collection of eukaryotic POL II promoters where the transcription start site (TSS) has been determined experimentally. The core promoter elements in the region were identified using the Elements Navigation Tool (ElemeNT) [45], a user-friendly, web-based interactive tool for predicting and displaying putative core promoter elements and their biologically relevant combinations. ElemeNT’s predictions are based on biologically functional core promoter elements and can infer core promoter compositions. ElemeNT does not assume prior knowledge of the actual TSS position and can annotate any given sequence.

The bidirectional promoter region identified the CpG islands and other epigenetic marks using the UCSC Genome Browser (http://genome.ucsc.edu/ accessed on 21 September 2021). The CpG islands were plotted using EMBOSS Cpgplot (https://www.ebi.ac.uk/Tools/seqstats/emboss_cpgplot accessed on 15 September 2021) [1,2,111,112].

### 4.2. Mutation Analysis in Different Types of Tumors

The cBioPortal web server (http://www.cbioportal.org accessed 30 November 2021) [34] is a comprehensive website that explores, visualizes, and analyzes multidimensional cancer genomics data. We used the “cancer types summary” module on the “TCGA PanCanAtlas” dataset available on the cBioPortal web server. Furthermore, the correlation between the genetic alteration of the two genes and their overall survival prognosis was explored in the “comparison” module. The prognostic values are presented with log-rank *p*-values.

### 4.3. Gene Expression Analysis

A differential gene expression analysis, to investigate the changes in expression of specific genes in tumor vs. normal samples, was performed using TIMER2.0 [32] and GEPIA2 [33] online tools.

#### 4.3.1. TIMER2.0

TIMER2.0 (Tumor Immune Estimation Resource, version 2; http://timer.cistrome.org/ accessed on 15 October 2021) is a comprehensive resource for systematically analyzing differential gene expression levels between tumor and adjacent normal tissues. We used “THAP9” and “THAP9-AS1” as the inputs in the “Gene_DE” module to evaluate the expression levels of the two genes in tumor tissue and adjacent normal tissues from 32 cancer types from TCGA [30]. “THAP9-AS1” was not available in TIMER2.0.

#### 4.3.2. GEPIA2

GEPIA2 (Gene Expression Profiling Interactive Analysis, version 2; http://gepia2.cancer-pku.cn/#analysis accessed on 16 October 2021) is an interactive web server for analyzing mRNA expression data from tumors and normal samples from TCGA (the Cancer Genome Atlas) and GTEx (Genotype–Tissue Expression) projects. We used the “expression analysis box plots” module of GEPIA2 to obtain box plots of THAP9 and THAP9-AS1 expression levels between tumor and normal tissues. We set the *p*-value cutoff as 0.01, the log2FC (fold change) cutoff as 1, and used the “match TCGA normal and GTEx data” option.

### 4.4. Prognostic Analysis of THAP9 and THAP9-AS1

The GEPIA2 webserver was used to explore the prognostic values of THAP9 and THAP9-AS1 in different types of tumors in TCGA. The “survival map” module of GEPIA2 was used to obtain the overall survival (OS) and disease-free survival (DFS) significance map data with cutoff-high (50%) and cutoff-low (50%) values to split the high-expression and low-expression cohorts. The survival data were visualized with hazard ratio, 95% confidence interval, and log-rank *p*-values.

### 4.5. Guilt by Association Analysis

#### 4.5.1. Construction of Weighted Gene Co-Expression Network

GBA (guilt by association) analysis [53] was used to identify co-expressing genes in each tumor and associated normal samples.

Sample Collection: We downloaded HTSeq-counts RNA-Seq data of 33 tumor types from TCGA (the Cancer Genome Atlas) database (https://portal.gdc.cancer.gov/ accessed on 20 October 2021) using the GDC Data Transfer Tool (https://gdc.cancer.gov/access-data/gdc-data-transfer-tool accessed on 20 October 2021). The row names of the downloaded HTSeq-count matrix were Ensembl gene identifiers, and the column names represented TCGA sample IDs.The dataset included the expression profiles of 60,483 genes from 11,094 patients. Constraints used to generate the manifest file to be used with GDC Data Transfer Tool are as follows: data category: transcriptome profiling; data type: gene expression quantification; experimental strategy: RNA-Seq; workflow type: HTSeq-counts; data format: txt; access: open; program: TCGA. Details about the samples can be found in Appendix A: Cancer types with less than 3 normal samples (rows highlighted in red)] were excluded; thus we finally analyzed 22 cancer types. 

Sample Preprocessing: The Ensembl gene IDs link to gene information in the Ensembl database [37]. The “org.Hs.eg.db” R Bioconductor package [111] was used to convert the Ensembl gene IDs to the gene symbols. Ensembl IDs that did not have an official gene symbol were dropped from the analysis. After filtering the samples, we were left with gene expression values of 34,125 genes across 11,094 samples.

WGCNA analysis: The gene expression values of each tumor and paired normal samples were subjected to the WGCNA, R Bioconductor package [35] for the weighted co-expression network construction. We used the “blockwiseModules” function in WGCNA, which performs automatic network construction and module detection processes on large expression datasets in a block-wise manner. In summary, it calculates the similarity matrix between each pair of genes across all samples based on its Pearson’s correlation value. Then, the similarity matrix is transformed into an adjacency matrix. Subsequently, the topological overlap matrix (TOM) and the corresponding dissimilarity (1-TOM) value are computed. Finally, a dynamic tree cut (DTC) algorithm detects gene co-expression modules. WGCNA distinguishes gene clusters by color names (tan, turquoise, brown etc.). The signed modules were constructed with a cut height of 0.995 and a minimum module size of 30 genes. Then, we used the modules associated with THAP9 and THAP9-AS1 in Cytoscape [112] to visualize the top 20 genes co-expressed with THAP9 and THAP9-AS1 in each tumor vs. normal pair.

#### 4.5.2. Gene Ontology (GO) and KEGG Pathway Enrichment Analysis

A GO analysis [113] is a helpful method for annotating genes and gene sets with biological characteristics for high-throughput genome or transcriptome data. The Kyoto Encyclopedia of Genes and Genomes (KEGG) pathway [114] is a knowledge base for the systematic analysis of gene functions. A GO and KEGG pathway enrichment analysis was performed using the “ShinyGO” web server [91], a Shiny application developed based on several R/Bioconductor packages. A *p*-value cutoff (FDR) of 0.05 was set as the cut-off criterion for extracting the top 10 enriched GO terms (biological process (BP), cellular component (CC), and molecular function (MF)) and KEGG pathways. Further, we merged all of the GO-BP, GO-CC, GO-MF, and KEGG pathways enriched in all tumor and normal samples and used the “word cloud” package in python to visualize the overall enrichment GO and KEGG pathways in normal vs. tumor samples.

#### 4.5.3. Differential Correlation Analysis

For the differential co-expression analysis between the tumor and normal samples in each cancer, R-package DGCA version 1.0.2 [36] was used. DGCA is an R package designed to detect differences in the correlations of gene pairs between distinct biological conditions. DGCA uses correlation coefficients transformed into normalized Z-scores to identify differentially correlated genes and modules while performing downstream analysis, including data visualization, GO enrichment, and network construction tools.

Firstly, we checked the correlation between THAP9 and THAP9-AS1 in each tumor vs. normal pair using the “plotCors” function in DGCA, which uses Pearson’s correlation by default. Following this, genes that were differentially correlated with THAP9 and THAP9-AS1 were computed with the “ddCorAll” function using “corrType” as the Spearman correlation. This pipeline provided the Spearman coefficient and the corresponding *p*-values for each pair of genes across samples. Significant changes in differential correlation between the two conditions (tumor vs. normal) were then identified using a Fisher’s Z-test. The correlation between THAP9/THAP9-AS1 and other genes was classified as having a gain of correlation or loss of correlation and based upon the threshold for correlation significance; the gene pairs were grouped into nine different correlation classes (+/+; +/−; +/0; −/+; −/0; −/−; 0/+; 0/0; 0/−). The classes show the correlation as positive (+), negative (−), or not significant (0) for each gene and condition when contrasting the groups (tumor/normal). A GO term enrichment analysis of differential correlation-classified genes was performed using the DGCA function “ddcorGO.”

## 5. Conclusions

We conducted a pan-cancer analysis of the THAP9 and THAP9-AS1 gene pair in various cancers. We explored the association of their aberrant expression with patient survival outcomes, followed by analyzing their functional association using gene co-expression and differential gene correlation analysis. This study has several limitations. Firstly, although we used TCGA and GTEx datasets, data about particular cancer types were not available. Secondly, given the myriad individual differences among cancer patients, it was challenging to cover all possible variations. Finally, our analysis was solely computational and relied on public databases. Future studies to validate the expression and function of the two genes at the cellular and molecular levels will shed more light on their physiological and pathological relevance.

## Figures and Tables

**Figure 1 ncrna-08-00051-f001:**
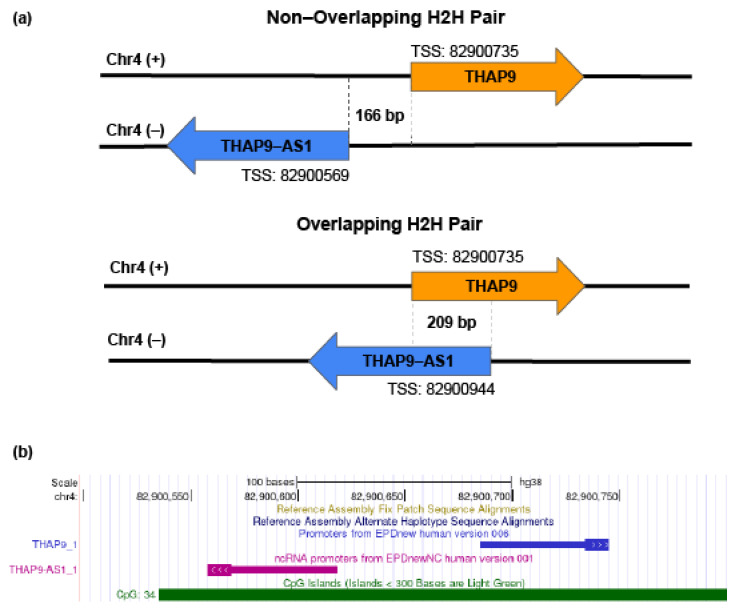
Identification of putative bidirectional THAP9/THAP9-AS1 promoter. (**a**) Schematic representation of the bidirectional genomic organization of THAP9 and THAP9-AS1 genes along with the TSS predicted by EPDnew. (**b**) UCSC genome browser showing THAP9 and THAP9-AS1 genes transcribed divergently based on the human GRCh38 assembly. CpG islands overlapping with the bidirectional promoter region are also indicated.

**Figure 2 ncrna-08-00051-f002:**
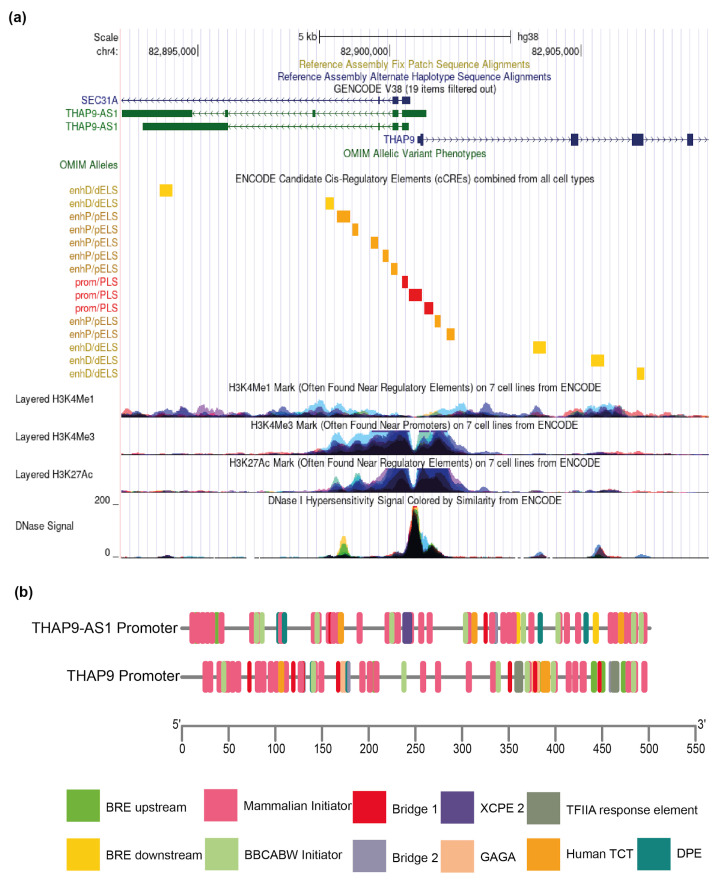
Characterization of THAP9/THAP9-AS1 putative bidirectional promoter region. (**a**) UCSC genome browser representing ENCODE data for THAP9/THAP9-AS1 bidirectional promoter region. The genomic region contains the putative bidirectional promoter region of the THAP9/THAP9-AS1 gene pair. The GENCODE genes track shows transcript variants for both genes. Below that is the ENCODE candidate cis-regulatory elements (cCREs) track, which shows the presence of several regulatory elements in the promoter region. The next three tracks are from ENCODE showing the H3K4Me1, H3K4Me3 and H3K27Ac marks followed by the DNAse I hypersensitivity signal shown in the last track (**b**) Schematic representation of the core promoter elements predicted by ElemeNT. The core promoter sequence used was −250 to +250 relative to the TSS of THAP9 and −400 to +100 for THAP9-AS1 (considering 82900569 as TSS) from EPDnew. The diagram is roughly to scale and was constructed using TBtools [43].

**Figure 3 ncrna-08-00051-f003:**
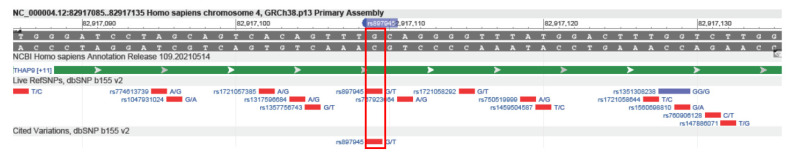
Genome Variation Viewer view of rs897945, which yields a G → T nucleotide substitution that leads to a Leu-to-Phe amino acid change at position 299 located on the Tnp_P_element (Pfam ID: PF12017) domain in hTHAP9 protein.

**Figure 4 ncrna-08-00051-f004:**
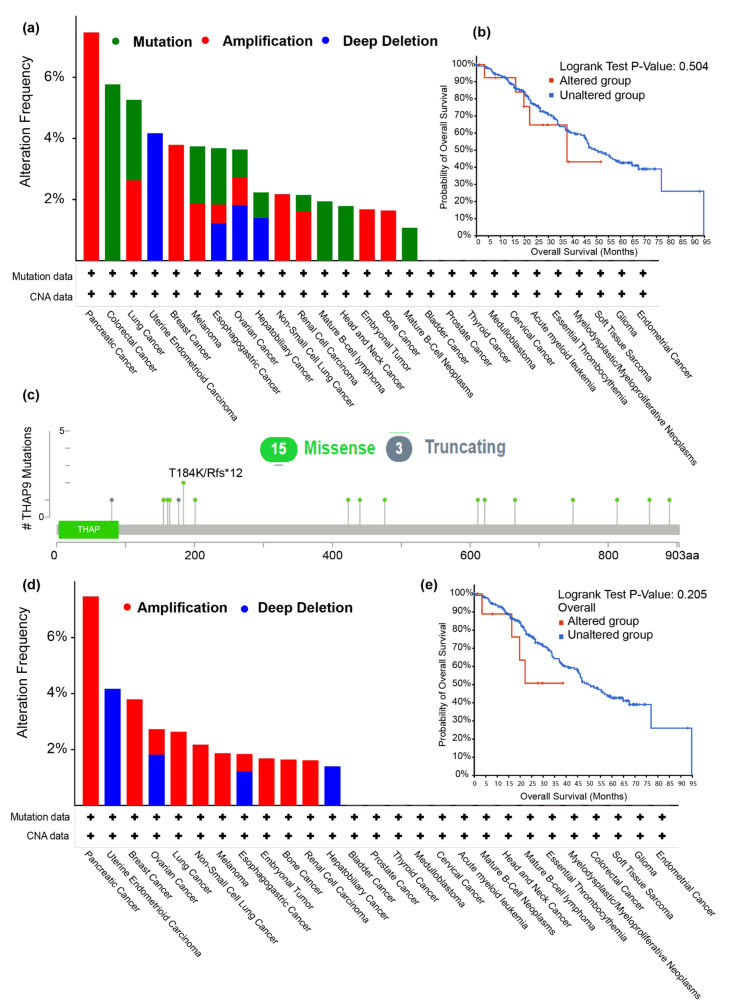
Mutations of THAP9 and THAP9-AS1 in different cancers in TCGA. The alteration frequencies with mutation type for (**a**) THAP9 and (**d**) THAP9-AS1, where the X-axis represents the type of alteration (red—amplification; blue—deep deletion; green—mutation) and the Y-axis represents the frequency of the alteration in different cancers. Correlation between mutation status and overall survival of cancer patients in (**b**) THAP9 and (**e**) THAP9-AS1. The red line shows the overall survival estimates for patients with an alteration in the gene as compared to patients with no alteration (blue line). Survival analysis significance was based on the log-rank test. Note: *p* < 0.05 was considered significant. (**c**) Mutation sites in THAP9 (refer to Appendix A for details).

**Figure 5 ncrna-08-00051-f005:**
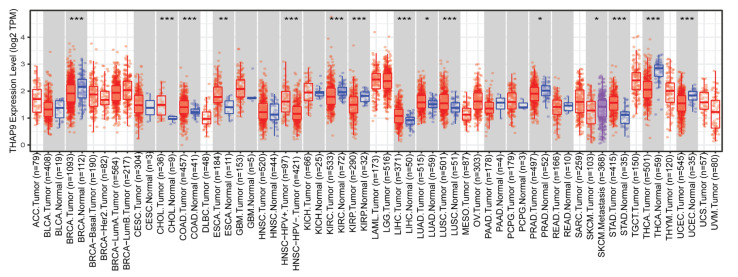
THAP9 gene expression levels in different tumors. THAP9 expression levels in human tumors (red) and corresponding normal tissues (blue) were obtained through TIMER2. The statistical significance computed by the Wilcoxon test is annotated by the number of stars (*: *p*-value < 0.05; **: *p*-value < 0.01; ***: *p*-value < 0.001).

**Figure 6 ncrna-08-00051-f006:**
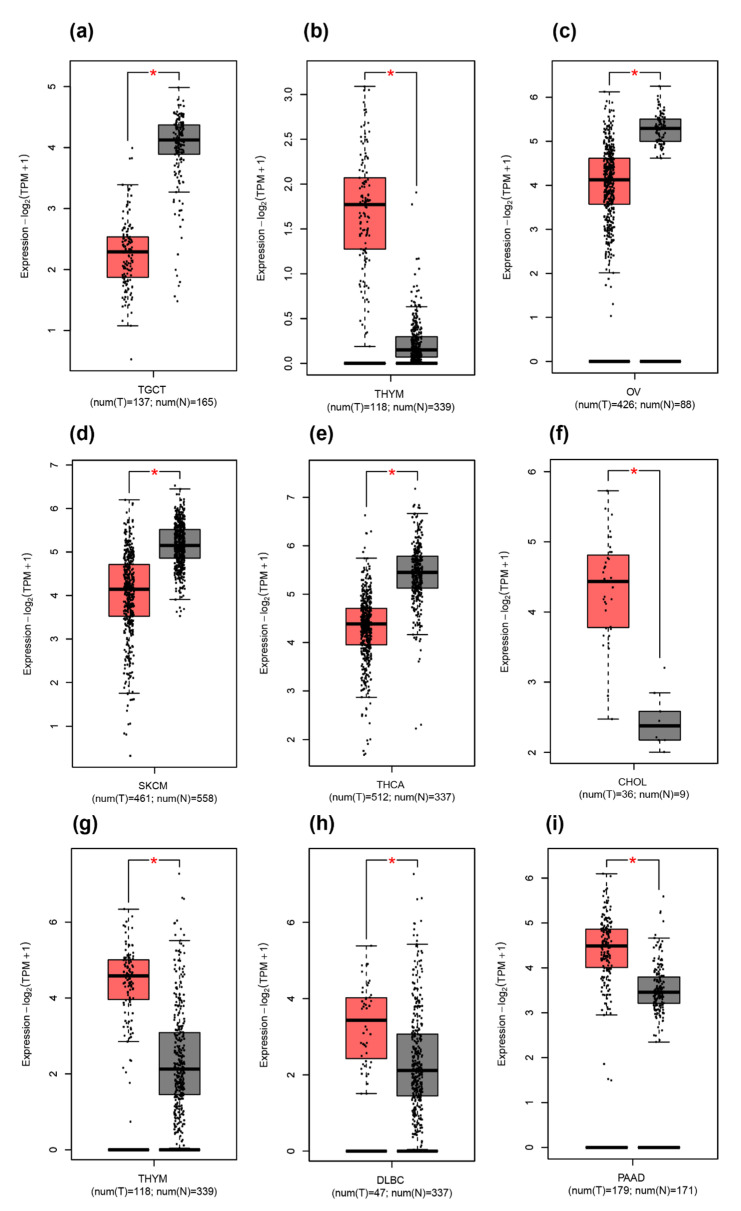
Box plot representation of the comparative expression levels of THAP9 (**a**,**b**) and THAP9-AS1 (**c**–**i**) in different tumor samples (red) vs. normal tissue samples (grey) from TCGA and GTEx generated using GEPIA2. Note: * *p* < 0.01. GEPIA2 uses one-way ANOVA, taking the pathological stage (X-axis) as the variable for performing differential expression of the input gene. The expression data used for the analysis was log2(TPM+1) (Y-axis)-transformed. (**a**) THAP9 is downregulated in TGCT and (**b**) upregulated in THYM. (**c**–**e**) THAP9-AS1 is downregulated in OV, SKCM, and THCA and (**f**–**i**) upregulated in CHOL, THYM, DLBC, and PAAD (* *p* < 0.05).

**Figure 7 ncrna-08-00051-f007:**
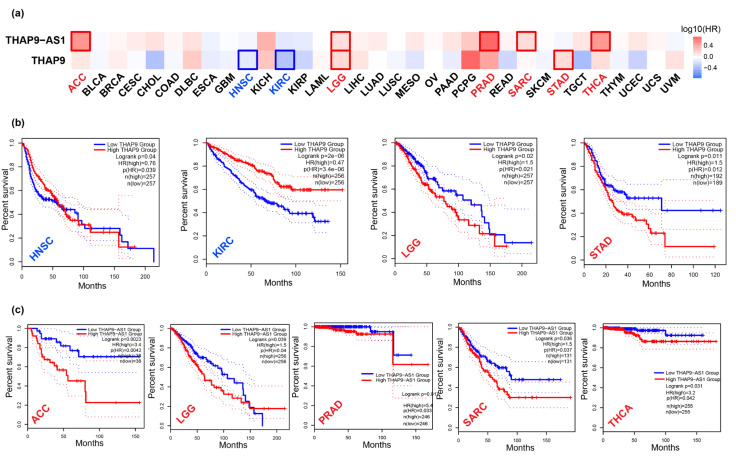
Overall patient survival analysis using GEPIA2. (**a**) The relationship between THAP9 and THAP9-AS1 gene expression and the overall survival prognosis of cancers in TCGA. Median was selected as a threshold for separating high-expression and low-expression cohorts. The red and blue blocks represent higher and lower risks, respectively, with an increase in gene expression. The bounding boxes depict the significant (*p* < 0.05) unfavorable and favorable results, respectively. The overall survival and gene expression rates (from TCGA) of (**b**) THAP9 in HNSC, KIRC, LGG, and STAD; and of (**c**) THAP9-AS1 in ACC, LGG, PRAD, SARC, and THCA.

**Figure 8 ncrna-08-00051-f008:**
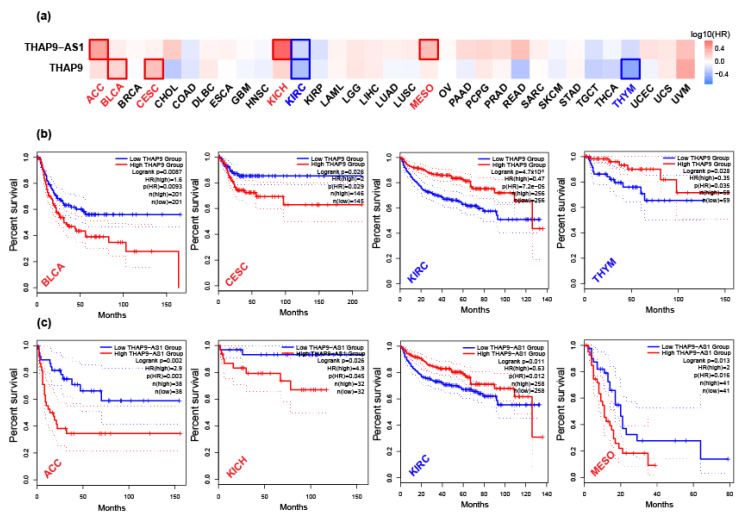
Disease-free survival analysis using GEPIA2. (**a**) The relationship between THAP9 and THAP9-AS1 gene expression and the disease-free survival prognosis of cancers in TCGA. The median was selected as a threshold for separating high-expression and low-expression cohorts. The red and blue blocks represent higher and lower risks, respectively, with an increase in the gene expression. The bounding boxes depict the significant (*p* < 0.05) unfavorable and favorable results, respectively. The disease-free survival and gene expression rates of (**b**) THAP9 in BLCA, CESC, KIRC, and THYM; and of (**c**) THAP9-AS1 in ACC, KICH, KIRC, and MESO.

**Figure 9 ncrna-08-00051-f009:**
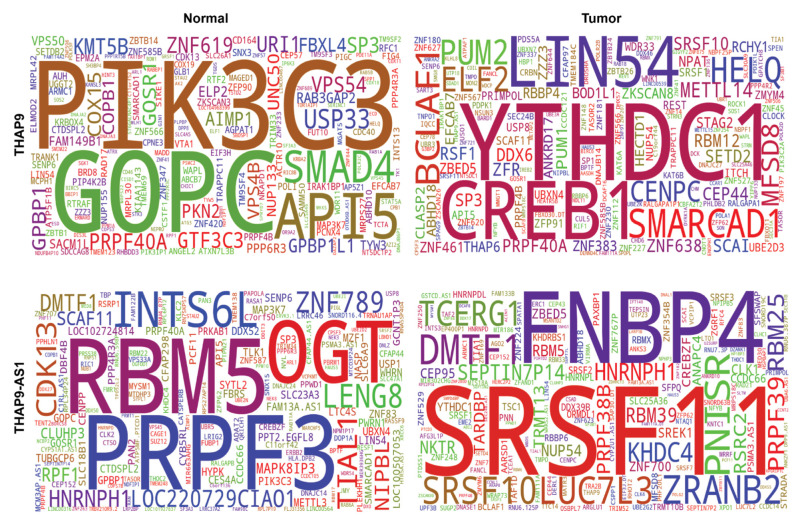
Consensus of genes co-expressed with THAP9 and THAP9-AS1. Word cloud of top 20 genes frequently co-expressed with: (1st row) THAP9 in all combined normal (**top-left**) vs. tumor (**top-right**) samples; (2nd row) THAP9-AS1 in combined normal (bottom-left) vs. tumor (**bottom-right**) samples. The co-expressing genes were identified using the WGCNA Bioconductor package and plotted using the Wordcloud python package. The height of a word is directly proportional to the frequency of co-expression with THAP9. The top 20 co-expressed genes are plotted only for representation purposes; details of the full gene cluster associated with THAP9 and THAP9-AS1 in each cancer type (normal and tumor tissues separately) are available in Appendix A.

**Figure 10 ncrna-08-00051-f010:**
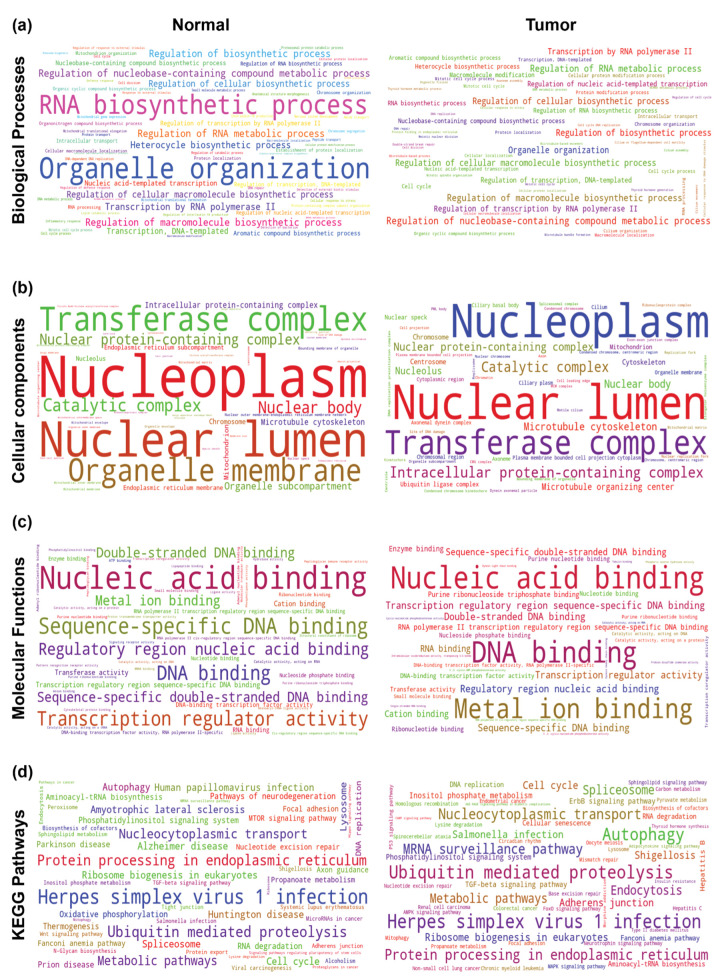
Gene Ontology (GO) and KEGG pathway analyses of genes co-expressed with THAP9 in normal vs. tumor samples. The enrichment test was performed for normal vs. tumor samples (for each cancer) using the ShinyGO for the top 10 enriched terms, with the significance cutoff for adjusted *p*-values bring set at 0.05. The font sizes in the word cloud are proportional to their frequency after the enrichment rates were merged for all cancers (left side for normal samples and right side for tumor samples). Word clouds of enriched GO terms in (**a**) the biological process category, (**b**) cellular component category, (**c**) molecular functions category, and (**d**) KEGG pathways.

**Figure 11 ncrna-08-00051-f011:**
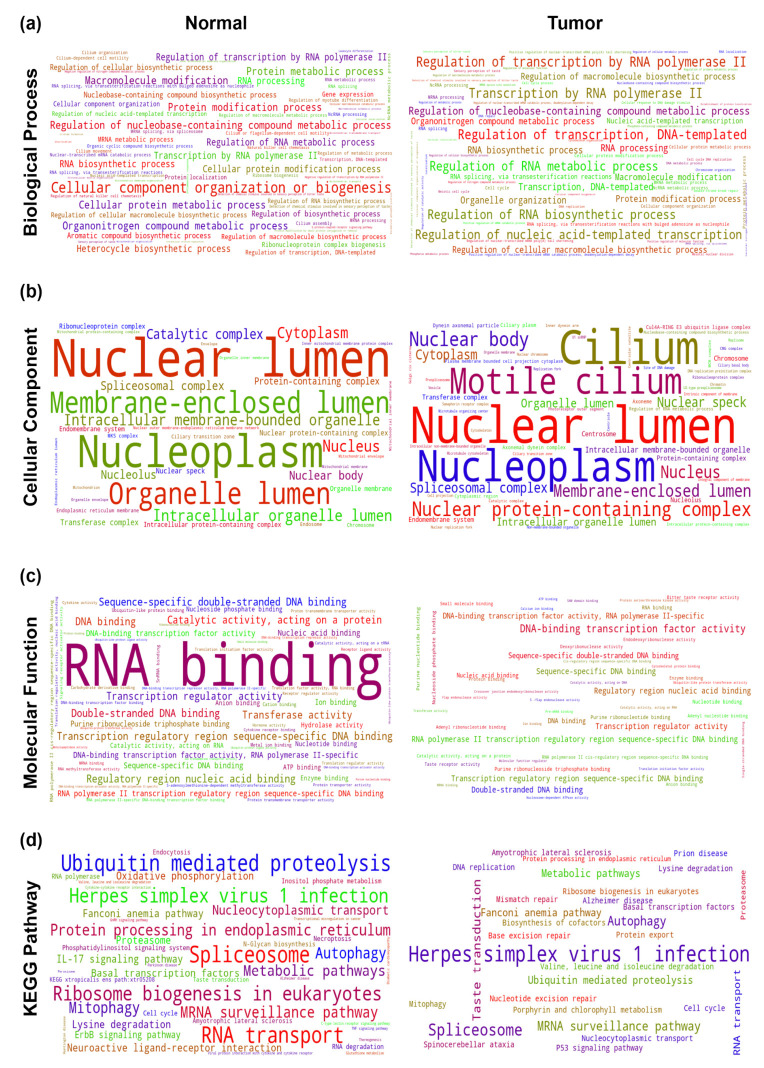
Gene ontology (GO) and KEGG pathway analyses of genes co-expressed with THAP9-AS1 in normal vs. tumor samples. The enrichment test was performed for normal vs. tumor samples (for each cancer) using ShinyGO for the top 10 enriched terms, with the significance cutoff for adjusted *p*-values being set at 0.05. The font sizes in the word cloud are proportional to their frequency after the enrichment rates were merged for all cancers (left side for normal samples and right side for tumor samples). Word clouds of enriched GO terms in (**a**) the biological process category, (**b**) cellular component category, (**c**) molecular functions category, and (**d**) KEGG pathways.

**Figure 12 ncrna-08-00051-f012:**
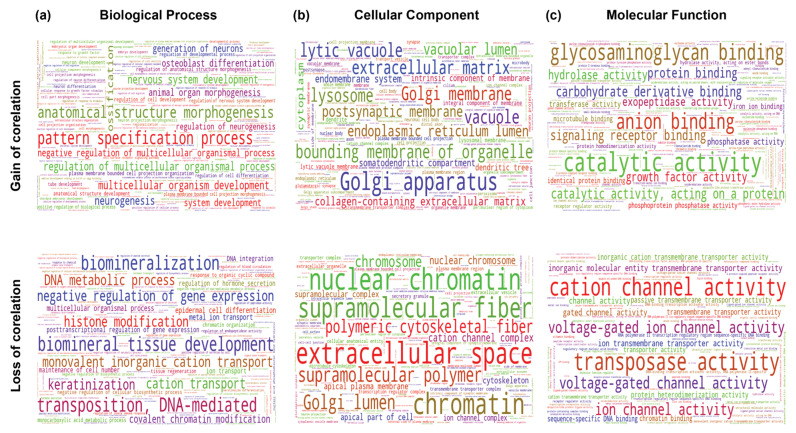
Gene Ontology analysis of genes differentially correlated with THAP9 in normal vs. tumor samples (top—genes that gained a correlation with THAP9; bottom—genes that lost a correlation with THAP9). Word clouds of enriched (**a**) GO biological process, (**b**) GO cellular component, and (**c**) GO molecular functions. The differential gene correlation analysis, followed by the gene ontology analysis for the differentially correlated genes, was performed using the DGCA package in R.

**Figure 13 ncrna-08-00051-f013:**
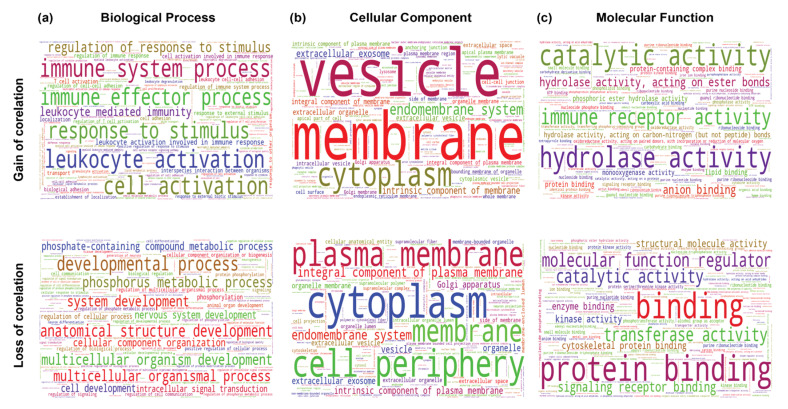
Gene Ontology analysis of genes differentially correlated with THAP9-AS1 in normal vs. tumor samples (top—genes that gained correlation with THAP9-AS1; bottom—genes that lost correlation with THAP9). Word clouds of enriched (**a**) GO biological process, (**b**) GO cellular component, and (**c**) GO molecular functions. The differential gene correlation analysis, followed by the gene ontology analysis for the differentially correlated genes, was performed using the DGCA package in R.

## Data Availability

Not applicable.

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
