# Peer review of "Pan-Cancer Analysis Reveals the Prognostic Potential of the THAP9/THAP9-AS1 Sense–Antisense Gene Pair in Human Cancers"

_ncrna, 2022, doi:10.3390/ncrna8040051_

Round 1
Reviewer 1 Report
Comments to the Author
The manuscript by Rashmi and Majumdar, systematically demonstrated that the expression of human THAP9 and THAP9-AS1 was strongly correlated with patient prognosis and higher expression of the gene pair was usually linked to poor overall and disease-free survival. It has been reported that THAP9 and THAP9-AS1 were involved in various cancers and the gene pair often co-express in normal and cancer samples. Interestingly, authors analyzed cancer datasets and found that the gene pair may serve as an important biomarker of tumor prognosis. Generally, the manuscript is well structured and contents are rich. The manuscript could be considered for publication after addressing the issues mentioned below.
Specific comments:
- In this study, authors explored the association of their co-expression of THAP9 and THAP9-AS1 with putative bidirectional promoter. However, if the gene pair was regulated by their putative bidirectional promoter, then why there are differential expression patterns in different cancer analyses. Please explain this conflict.
- The authors analyzed the expression, prognosis, and biological function of THAP9 and THAP9-AS1 mainly through cancer datasets (TCGA, GTEx), whether representative data were covered to ensure the accuracy of the analysis results.
- The manuscript repeatedly mentions that THAP9-AS1 is a newly annotated lncRNA, but there is barely introduced the background and functions of lncRNA.
- In this study, authors observed that both the THAP9 and THAP9-AS1 genes were mutated in several cancers, what implications for further studies?
- Please ensure that the reference style precisely.
- There is an overlap of figure and figure notes in Figure 6c
Author Response
Thanks for an encouraging review. Here is a point-by-point response to your comments.
1. In this study, the authors explored the association of their co-expression of THAP9 and THAP9-AS1 with putative bidirectional promoters. However, if the gene pair was regulated by their putative bidirectional promoter, then why are there differential expression patterns in different cancer analyses? Please explain this conflict.
Genes regulated by bidirectional promoters are usually positively or negatively correlated with each other. But there have been reports of conditional regulation of bidirectional gene pairs as well. For example, some gene pairs like murine RanBP1/Htf9-c are coregulated only in a common window of the cell cycle [7,52]. On the other hand, human HSP60/HSP10 display coordinated expression in response to induction signals [8]. This can explain why THAP9 and THAP9-AS1 show differential expression patterns in different cancer analyses. THAP9-AS1 is consistently upregulated under stress, whereas THAP9 exhibits both downregulation and upregulation [29].
However, it is to be noted that when we compared the expression patterns of the two genes, THAP9 and THAP9-AS1, with each other, they always showed positive correlation.
Also explained in Section 2.3 (pg 10, Line 215 onwards highlighted text)
2. The authors analyzed the expression, prognosis, and biological function of THAP9 and THAP9-AS1 mainly through cancer datasets (TCGA, GTEx), whether representative data were covered to ensure the accuracy of the analysis results.
We have used datasets curated by ENCODE, EPD, TCGA and GTEx mostly via previously published, highly used tools like UCSC genome browser, ElemeNT, TIMER2, GEPIA2 etc. Only for performing the “Guilt by association analysis” we downloaded and processed the data ourselves.
Details about the data and the processing can be found in Section 4.5.1 (pg 20).
3. The manuscript repeatedly mentions that THAP9-AS1 is a newly annotated lncRNA, but there is barely introduced the background and functions of lncRNA.
Added to Introduction section (pg 2 Line 53 onwards, highlighted text): THAP9-AS1 is a lncRNA gene located on chromosome 4q21.22. It was reported to be upregulated in nasopharyngeal carcinoma and breast cancer [23,24]. Further it was also identified as an oncogenic factor promoting cell growth in pancreatic ductal adenocarcinoma and gastric cancer and played a role in the apoptosis of spontaneous neutrophils [25–27]. The THAP9-AS1 knockdown was observed to suppress cell proliferation and enhance apoptosis in esophageal squamous cell carcinoma (ESCC) [28]. A recent study from our lab reported that THAP9 and THAP9-AS1 exhibit different gene expression patterns under different types of stresses in the S-phase of the cell cycle. THAP9-AS1 is consistently upregulated under stress, whereas THAP9 exhibits both downregulation and upregulation in various stress conditions. Both THAP9 and THAP9-AS1 exhibit a periodic gene expression throughout the S-phase, which is a characteristic of cell cycle-regulated genes [29].
4. In this study, authors observed that both the THAP9 and THAP9-AS1 genes were mutated in several cancers, what implications for further studies?
We made certain interesting observations from the mutation studies. For instance, in cancers like pancreatic cancer (THAP9-AS1 has been implicated in pancreatic cancer), breast cancer, non-small cell lung cancer, melanoma, embryonal tumor, and bone cancer, where both the genes show similar mutation profiles (“amplification” i.e., more copies, often focal), it will be interesting to further investigate the co-regulation of the THAP9-THAP9-AS1 putative head-to-head gene pair. We also observed that many of the mutations lie in the C-Terminal end of the THAP9 protein which is largely unexplored: we plan to experimentally validate the role of these mutations in cancer progression.
5. Please ensure that the reference style precisely.
Corrected.
6. There is an overlap of figure and figure notes in Figure 6c
Corrected. Please check Figure 7.
Reviewer 2 Report
The current manuscript by Rashmi & Majumdar describes pan-cancer analysis of THAP9 and THAP9-AS1 gene pair using publicly available cancer datasets and predicted a bidirectional promoter that regulates the expression. Authors have provided several figures and reviewer felt the connection is missing. In some results, there is no correlation between both the genes expression and some datasets show correlation in THYM. There is no experimental validation for any of the predictions/hypothesis based on reanalysis of TCGA/GEPIA datasets. The detailed comments are as follows:
- The bidirectional promoter predicted by EPDnew analysis and Encode histone marks (in seven cell lines) has not validated. Does this regulation is constitutively active or specific to some cancers? The Encode data in Figure-2 just shows 7 cell lines data only. What are those cell lines (all cancer or any normal cells)? The data from downstream analysis do not show such bidirectional regulation of gene pair in any cancer except THYM (Figure-5).
- Authors represented alteration frequency in Figure-3 for two genes separately. It would be better to compare together to see the frequencies for two gene pair and correlations if nay.
- Figure-4 data is not making any sense. 4A describes significant expression changes in few cancer types and the same data from TCGA analyzed by EdgeR shows no expression of either gene without any correlation between the gene pair.
- Authors should be aware before using TCGA data for analysis. As depicted in 4B, TCGA datasets have very less or NO normal control samples for analysis of such differential expression. In results, authors used tumor abbreviations throughout without expanding. Though the abbreviations are standard from TCGA/GEPIA, it would be better for readers to understand what those abbreviations (at least the cancer type).
- Please elaborate in Figure-4 legend about p-values and what each of symbols mean (*/**/***).
- In the abstract, authors indicated that “higher expression of gene pair was linked to poor overall survival…”. But, Figure-5 depicts down/upregulation of THAP9-AS1 in most of cancer types. These data in accordance with 4C, shows no coregulation of gene pair through same promoter in any cancer other than THYM as authors stated. Then so many questions on overall conclusions.
- The cancer types discussed and highlighted in Figure-5 were different from Figure-6 &7. The expression of these genes were not significant when compared to normal control in Figure-5, while authors could see HR in other cancers. Not really connected? Also, the heatmap shows differential regulation of both THAP9 and its anti-sense gene without indicating any bidirectional regulation except in few cancer types.
- Figures 8-12, it is not clear tumor means (any specific type). Just top 20 genes common in all cancer types were analyzed? If authors used paired tissues (normal and tumor) from TCGA, very few datasets have matched controls. GEPIA normal do not match with tumors from TCGA. SO, make it clear which datasets were used for Figures 8-12. The analysis of tumor and normal separately in 8-12 figures indicate in terms of significance of gene pair in cancer. The word cloud of 20 genes is just very hard to read. Authors could show a bar graph or list in table or bar graph.
- Such analysis with top-20 genes only do not make overall regulatory roles of these genes in cancers.
- Having all the cancer types with different expressions varying from dataset to dataset, it is very hard to agree with author conclusions.
- Majorly, the manuscript has no experimental validation of any of conclusions that were based on small set of (top-20) genes and inconsistent tumor data related to expression or correlation on co-expressing genes.
- Also, the genes identified (top-20) to be co-express with THAP9 gene pair definitely needs validation of their role in connection with TAHP9 functions. It is good that authors were able to hypothesize their co-expression with target genes, but functional validation is still missing.
- It is good to start with pan cancer analysis of THAP9 & its anti-sense gene, but it is better to narrow down to a particular cancer to elucidate co-expressional regulation of gene pair studied in this manuscript and adding few experimental validations to prove the conclusions.
- Manuscript definitely needs attention for language editing. Some of the statements are hard to understand.
Minor:
1) please add line numbers in order to point out specific sentences for editing/review purpose.
2) The fourth paragraph on page-2, authors wrote few sentences on THAP9 gene pair expression in stress conditions. They are not clear for reading and hard to understand the authors intention.
3) There is lot of new literature around the significant role of THAP9-AS1 in esophageal squamous cell carcinoma (ESCC), osteosarcoma (Yang S eta l. 2021), stem cell regulation (Wang J. et al. 2021) and citing them in discussion or introduction would be better.
4) It is not clear how much important the genes that gained or lost correlation with that of THAP9 gene pair.
5) The word cloud figures are hard to read few genes or processes or pathways due to over congested words. Authors should present data which is easy to read and understand by common readers.
Best wishes,
Author Response
Thanks for the detailed review. Here is a point-by-point response to your comments.
1. The bidirectional promoter predicted by EPDnew analysis and Encode histone marks (in seven cell lines) has not validated. Does this regulation is constitutively active or specific to some cancers? The Encode data in Figure-2 just shows 7 cell lines data only. What are those cell lines (all cancer or any normal cells)? The data from downstream analysis do not show such bidirectional regulation of gene pair in any cancer except THYM (Figure-5).
The 7 cell lines included in this track are GM12878 (lymphoblastoid cell line), H1-hESC (Embryonic stem cell line derived from human blastocysts), HSMM (Human Skeletal Muscle Myoblasts), HUVEC (Human umbilical vein endothelial cells), K562 (human immortalized myelogenous leukemia cell line), NHEK (Primary Human Keratinocytes), NHLF (Human Lung Fibroblasts).
Details about the ENCODE tracks can be found in Section 2.1 (pg 4, Line 111 onwards). We acknowledge and speculate about why THAP9 and THAP9-AS1 do not coregulate in many cancers in Section 2.3 (pg 10, Line 215 onwards, highlighted text).
2. Authors represented alteration frequency in Figure-3 for two genes separately. It would be better to compare together to see the frequencies for two gene pair and correlations if any.
We have made that comparison in Section 2.2 (Pg 6, Line 161 onwards). The data obtained from cbioportal, gives the frequency and percentage of mutations in individual genes, not the actual count/raw data. Thus, we have made comparisons based on cancer or mutation types.
3. Figure-4 data is not making any sense. 4A describes significant expression changes in few cancer types and the same data from TCGA analyzed by EdgeR shows no expression of either gene without any correlation between the gene pair.
Corrected. Please check Figure 5. We have removed the EdgeR section because of missing datasets in many cancers (Please visit section 2.3, Pg 8).
4. Authors should be aware before using TCGA data for analysis. As depicted in 4B, TCGA datasets have very less or NO normal control samples for analysis of such differential expression. In results, authors used tumor abbreviations throughout without expanding. Though the abbreviations are standard from TCGA/GEPIA, it would be better for readers to understand what those abbreviations (at least the cancer type).
We have removed that section (related to previous Fig.4B) and replaced with Figure 5. Also added the abbreviations section (Pg 21).
5. Please elaborate in Figure-4 legend about p-values and what each of symbols mean (*/**/***).
Added. Please refer to Figure 5 legend.
6. In the abstract, authors indicated that “higher expression of gene pair was linked to poor overall survival…”. But, Figure-5 depicts down/upregulation of THAP9-AS1 in most of cancer types. These data in accordance with 4C, shows no coregulation of gene pair through same promoter in any cancer other than THYM as authors stated. Then so many questions on overall conclusions.
We have modified this Section and removed the part associated with EdgeR due to inconsistencies in the datasets used. Please refer to section 2.3 (pg 8).
After combining the results from the TIMER2 and GEPIA2 methods, we observed that THAP9 and THAP9-AS1 expressions were coordinately upregulated in CHOL and THYM and were coordinately downregulated in THCA compared with the corresponding normal samples (Fig.6, Supplementary Figure 5 and Supplementary Figure 6).
Head-to-Head genes are often coregulated by bidirectional promoters, although there have been reports of conditional regulation of bidirectional gene pairs as well. For example, some gene pairs like murine RanBP1/Htf9-c are coregulated only in a common window of the cell cycle [7,52]. On the other hand, human HSP60/HSP10 display coordinated expression in response to induction signals [8]. We have previously reported that THAP9 and THAP9-AS1 exhibit different gene expression patterns under diverse stress conditions in the S-phase of the cell cycle. THAP9-AS1 is consistently upregulated under stress, whereas THAP9 exhibits both downregulation and upregulation [29].
Thus, given the above, it is possible that THAP9 and THAP9-AS1 show diverse expression patterns in different cancers. However, it is to be noted that when we compared the expression patterns of the two genes with each other, they always showed a positive correlation. The differential expression of THAP9 and THAP9-AS1 in different tumor types suggests that the two genes may have tumor-specific regulatory mechanisms.
7. The cancer types discussed and highlighted in Figure-5 were different from Figure-6 &7. The expression of these genes were not significant when compared to normal control in Figure-5, while authors could see HR in other cancers. Not really connected? Also, the heatmap shows differential regulation of both THAP9 and its anti-sense gene without indicating any bidirectional regulation except in few cancer types.
The heatmaps in Fig 7, 8 are comparing the HR i.e. higher and lower risks, respectively with an increase in gene expression; not expression levels. Also as explained in Section 2.3, the two genes do not show similar expression patterns in all cancers possibly because of alternative coregulation by bidirectional promoters, such that there is a condition-specific coregulation of the two genes. In our future studies we will design experiments to study this in further detail.
8. Figures 8-12, it is not clear tumor means (any specific type). Just top 20 genes common in all cancer types were analyzed? If authors used paired tissues (normal and tumor) from TCGA, very few datasets have matched controls. GEPIA normal do not match with tumors from TCGA. SO, make it clear which datasets were used for Figures 8-12. The analysis of tumor and normal separately in 8-12 figures indicate in terms of significance of gene pair in cancer. The word cloud of 20 genes is just very hard to read. Authors could show a bar graph or list in table or bar graph.
Please visit Section 4.5.1 (Methods) and Supplementary table 3 for details about the datasets used. We have used only the TCGA data for the Guilt by association analysis and left out the cancers which had less than 3 paired normal samples. Figures 8-10 are just for representation purposes. We have done the analysis for the full set of genes and not just top 20: please visit supplementary tables 7-12 for full datasets.
9. Such analysis with top-20 genes only do not make overall regulatory roles of these genes in cancers.
We have performed this analysis for all the genes coexpressing with THAP9 and THAP9-AS1 (Supplementary tables 7-12). Top 20 genes have been used just for representation purposes in the main text; otherwise it would be difficult to represent the data. Our goal was to illustrate the enriched ontology and pathways associated with genes that show high correlation with THAP9 expression (Figure 10-1). Future experiments will test the association between THAP9 and enriched gene ontologies and pathways. It is to be noted that the roles of THAP9 and THAP9-AS1 in humans are not known. This analysis opens new avenues for further studies associated with these genes.
10. Having all the cancer types with different expressions varying from dataset to dataset, it is very hard to agree with author conclusions.
GEPIA2 takes care of the differences between the TCGA and GTEX datasets during data integration. For GBA, we only used TCGA datasets.
11. Majorly, the manuscript has no experimental validation of any of conclusions that were based on small set of (top-20) genes and inconsistent tumor data related to expression or correlation on co-expressing genes.
Human THAP9, that encodes a domesticated transposase of unknown function and lncRNA THAP9-AS1 (THAP9 -antisense1) are arranged head-to-head on opposite DNA strands forming a sense and antisense gene pair, which are possibly regulated by a bidirectional promoter. It is to be noted that the roles of THAP9 and THAP9-AS1 are not known. Although both THAP9 and THAP9-AS1 are reported to be involved in various cancers, their correlative roles on each other’s expression has also not been explored.
This study has adopted solely computational methods, to analyse the vast amounts of publicly available clinical data, and possibly decipher the physiological as well as pathological roles of THAP9 and THAP9-AS1. The co-expressed genes/pathways identified in this study will help us design experiments to study the role of these genes under specific conditions (e.g., certain cancers or neurodegenerative disorders, cellular processes). This computational analysis opens new avenues for further experimental studies about these genes.
12. Also, the genes identified (top-20) to be co-express with THAP9 gene pair definitely needs validation of their role in connection with TAHP9 functions. It is good that authors were able to hypothesize their co-expression with target genes, but functional validation is still missing.
We have performed this analysis for all the genes coexpressing with THAP9 and THAP9-AS1 (Supplementary tables 7-12). Top 20 genes have been used just for representation purposes in the main text. Also see comment above: This purely computational analysis opens new avenues for further experimental studies about these genes.
13. It is good to start with pan cancer analysis of THAP9 & its anti-sense gene, but it is better to narrow down to a particular cancer to elucidate co-expressional regulation of gene pair studied in this manuscript and adding few experimental validations to prove the conclusions.
We completely agree with the reviewer. This purely computational analysis opens new avenues for further experimental studies about these genes and their involvement in disease (e.g., certain cancers or neurodegenerative disorders) and/or cellular processes.
14. Manuscript definitely needs attention for language editing. Some of the statements are hard to understand.
Have re-written some sections as per reviewer comments.
Minor:
1) please add line numbers in order to point out specific sentences for editing/review purpose.
Line numbers have been added.
2) The fourth paragraph on page-2, authors wrote few sentences on THAP9 gene pair expression in stress conditions. They are not clear for reading and hard to understand the authors intention.
This was added to highlight a previous study (ref. 29) which suggests that the THAP9 and THAP9-AS1 genes can be conditionally coregulated. Have rewritten this paragraph.
3) There is lot of new literature around the significant role of THAP9-AS1 in esophageal squamous cell carcinoma (ESCC), osteosarcoma (Yang S eta l. 2021), stem cell regulation (Wang J. et al. 2021) and citing them in discussion or introduction would be better.
Thanks for this comment. Added these references.
4) It is not clear how much important the genes that gained or lost correlation with that of THAP9 gene pair.
It is suggested that if there is a change in the correlation between the expression of two genes under certain conditions (i.e., they are differentially correlated), they possibly regulate or are regulated by the condition [93–95]. Many studies have used differential correlation analyses to identify genes underlying differences between healthy and disease samples or between different tissues, cell types, or species [96–99]. Genes that are functionally related tend to have similar expression profiles; therefore, differential gene correlation analysis that can compare the expression correlation of THAP9 and THAP9-AS1 with other genes in normal vs. tumor samples can give us insight into biological processes and molecular pathways that distinctly involves the two genes in the two conditions.
The genes identified by this study will help us in designing experiments for studying the role of THAP9 in specific conditions (e.g., certain cancers or neurodegenerative disorders, cellular processes).
5) The word cloud figures are hard to read few genes or processes or pathways due to over congested words. Authors should present data which is easy to read and understand by common readers.
Supplementary Tables 7-12 have detailed information, using which the word clouds were created. We can include these in the main text, if recommended.
Reviewer 3 Report
The authors performed a comprehensive bioinformatic analysis of the THAP9 and THAP9-AS1 head to head gene pair in multiple cancers. Although both THAP9 and THAP9-AS1 are reported to be involved in various cancers, their correlative roles on each other’s expression has not been explored. They observed that although the expression of the two genes, THAP9 and THAP9-AS1, varied in different tumors, the expression of the gene pair was strongly correlated with patient prognosis. Thus, THAP9 and THAP9-AS1 may serve as a potential clinical biomarker of tumor prognosis. Further, they reported that in both normal and cancer samples, THAP9 and THAP9-AS1 often co-express; moreover, their expression is positively correlated in each cancer type suggesting the coordinated regulation of this H2H gene pair. I think these findings are interesting. I have a few minor comments below.
- Some figures are not mentioned in the main text such as Fig 3e.
- The p-value is not less than 0.05 in Fig 3b and 3e; can you conclude that both THAP9 and THAP9-AS1 alterations reflected poor overall survival and prognosis in cancer.
- Full names were not given for many abbreviations appearing for the first time.
Author Response
Thanks for an encouraging review. Here is a point-by-point response to your comments.
1. Some figures are not mentioned in the main text such as Fig 3e.
Corrected, please visit Section 2.2 (pg 6).
2. The p-value is not less than 0.05 in Fig 3b and 3e; can you conclude that both THAP9 and THAP9-AS1 alterations reflected poor overall survival and prognosis in cancer.
Yes, if we look at Figure 4b and 4e, the P values for both THAP9 and THAP9-AS1 are not less than 0.05, rather it is 0.2 for THAP9 and 0.5 for THAP9-AS1 which implies that the alterations may not reflect a significant impact on the poor overall survival and prognosis in cancer. This also correlates with the alteration frequency displayed in Figure 4a and 4d, i.e., none of the alterations show more than 6% frequency. Added on pg 6, Line 171 highlighted text).
3. Full names were not given for many abbreviations appearing for the first time.
Abbreviations section added on pg 21.
Round 2
Reviewer 2 Report
Dear Authors,
Thanks for providing the revised version and addressing most of the reviewer suggestions. Overall the computational analysis presented in this paper would encourage future studies to validate the role of gene pair in cancers.
Few minor suggestion:
1) It would be good to change 'human cancer' to 'human cancers' in the TITLE as the papers discuss role of gene pair in various cancers.
2) In line 155, end the sentence after cell lines with '.'
3) In line 163, change 'H3K4me3' to 'H3K4Me3' in consistent with other text.
4) In line 166, remove extra '.' at the end.
5) Figure 1C was cited and discussed after figure 2 in the section 2.1. You can move Figure 1C as 'Figure 2B' to follow the guidelines.
6) In section 2.2, please provide the allele frequency for rs897945. Is it 5% or more or less?
7) In figure-5 legend, please indicate the statistical package used for P-value computation. Is it ANOVA or Limma? In Figure-6, ANOVA was used.
8) In line 545, please cite the paper (
for EMBOSS Cpgplot tool used.9) Follow consistency with the use of 'coexpression' or as 'co-expression'. Authors used mix of these forms.
10) In lines 455-458, Figure 13 corresponds to THAP9-AS1. But the Figure-13 legend (line-460) still indicates as 'THAP9'. Correct it.
11) In line 454 & 464, please clarify 'DGCA R' as 'DGCA package in R' or 'R package DGCA'.
12) Please provide legends for supplementary figure & Tables in 'Supplementary materials' section in line '692' to give a brief idea to readers about supplementary data. Please use '.zip' version for as not every other readers familiar with '.7z' compression.
Best wishes,
Author Response
Thanks so much for the detailed review. Here is our point-by-point response:
1) It would be good to change 'human cancer' to 'human cancers' in the TITLE as the papers discuss role of gene pair in various cancers.
Corrected as per reviewers’ suggestion
2) In line 155, end the sentence after cell lines with '.'
Could not find this sentence.
3) In line 163, change 'H3K4me3' to 'H3K4Me3' in consistent with other text.
Corrected as per reviewers’ suggestion
4) In line 166, remove extra '.' at the end.
Corrected as per reviewers’ suggestion
5) Figure 1C was cited and discussed after figure 2 in the section 2.1. You can move Figure 1C as 'Figure 2B' to follow the guidelines.
Corrected as per reviewers’ suggestion
6) In section 2.2, please provide the allele frequency for rs897945. Is it 5% or more or less?
Already provided in text: “"The only reported missense SNP in THAP9 that has a global MAF > 0.05 (minor allele frequency) was rs897945 (MAF = 0.3253), making it a candidate for a derived allele (DA).”
7) In figure-5 legend, please indicate the statistical package used for P-value computation. Is it ANOVA or Limma? In Figure-6, ANOVA was used.
Added: The statistical significance computed by the Wilcoxon test is annotated by the number of stars (*: p-value < 0.05; **: p-value <0.01; ***: p-value <0.001).
8) In line 545, please cite the paper (DOI: 10.1093/nar/gkac240 PMID: 35412617) for EMBOSS Cpgplot tool used.
Added reference
9) Follow consistency with the use of 'coexpression' or as 'co-expression'. Authors used mix of these forms.
Corrected, changed everything uniformly to "co-expression" or "co-expressed"
10) In lines 455-458, Figure 13 corresponds to THAP9-AS1. But the Figure-13 legend (line-460) still indicates as 'THAP9'. Correct it.
Corrected
11) In line 454 & 464, please clarify 'DGCA R' as 'DGCA package in R' or 'R package DGCA'.
Corrected
12) Please provide legends for supplementary figure & Tables in 'Supplementary materials' section in line '692' to give a brief idea to readers about supplementary data. Please use '.zip' version for as not every other readers familiar with '.7z' compression.
Added as suggested by reviewer.